# Availability and readiness of health facilities for non-communicable disease services in Ethiopia: Evidence from the nationally representative health facility survey 2022

Andualem Yalew Aschalew[1]*, Jenberu Mekurianew Kelkay[2], Getachew Teshale[1], Kaleb Assegid Demissie[1], Nebebe Demis Baykemagn[3], Azmeraw Tadele[4], Misganaw Guadie Tiruneh[1], Tesfahun Zemene Tafere[1], Asebe Hagos[1], Melak Jejaw[1]

1 Department of Health Systems and Policy, Institute of Public Health, College of Medicine and Health Science, University of Gondar, Gondar, Ethiopia, 2 Department of Public Health, College of Health Science, Debark University, Debark, Ethiopia, 3 Department of Health Informatics, Institute of Public Health, College of Medicine and Health Science, University of Gondar, Gondar, Ethiopia, 4 Department of Medical Nursing, School of Nursing, College of Medicine and Health Science, University of Gondar, Gondar, Ethiopia

* yalewandualem@gmail.com

## Abstract

### Background

In response to the escalating non-communicable disease (NCD) challenge, effective management of NCDs requires sustained investments in infrastructure, trained health workforce, and the consistent availability of guidelines. Therefore, this study assessed both the availability and readiness of health facilities (HFs) to provide NCD-related services, while also examining how facility characteristics and health system factors are associated with service readiness.

### Methods

We analysed data from the nationally representative Ethiopia Service Provision Assessment (ESPA) survey 2021−22 to determine the availability and readiness of HFs for cardiovascular diseases (CVDs), diabetes, chronic respiratory diseases (CRDs) and mental, neurological and substance (MNS) use disorders-related services using the WHO Service Availability and Readiness Assessment manual. Readiness score was measured as the average availability of tracer items in percent, and HFs were considered 'ready' for NCDs management if they scored ≥70 (out of 100). We performed weighted multivariable logistic regression to identify factors affecting NCD service readiness.

**Data availability statement:** The dataset has been uploaded to Figshare, a public data repository. The dataset can be accessed via the DOI link: https://doi.org/10.6084/m9.figshare.30517046.

**Funding:** The author(s) received no specific funding for this work.

**Competing interests:** The authors have declared that no competing interests exist.

## Results

Approximately 93% reported offering services for diabetes, CVDs, and CRDs, while only 26% provided MH services. Overall service readiness remains critically low when applying the 70% readiness threshold. Only 15.68% of facilities were deemed ready for diabetes, 10.64% for CVDs, 3.14% for CRDs, and 11.52% for MNS use disorders-related services. Public facilities demonstrated significantly higher preparedness than private facilities. A number of factors have been associated with better service readiness, including the location of the facility, the level of the facility, having a quality control unit, conducting regular administrative meetings and receiving external supervision.

## Conclusions

This study reveals that overall service readiness for NCDs remains significantly low across HFs in Ethiopia. Public facilities and facilities located in urban settings demonstrated significantly higher levels of service readiness, highlighting substantial disparities in resource availability, integration with national health programs, and access to support systems. These findings underscore the need for Ethiopia's health system to move beyond service availability and focus on enhancing the preparedness of care delivery, with particular emphasis on equity and integration.

## Introduction

Noncommunicable diseases (NCDs) killed at least 43 million people in 2021. Of all NCD deaths, 73% were in low- and middle-income countries (LMICs). The four major NCDs: Cardiovascular diseases (CVDs), cancers, chronic respiratory diseases (CRDs) and diabetes, account for 80% of all premature NCD deaths [1].

In 2019, Ethiopia's estimated number of NCD deaths was 219,284, making a death rate of 204 per 100,000. CVDs, cancers, CRDs, and diabetes were found to be among the leading causes of death rates. In addition, mental, neurological and substance (MNS) use disorders become prevalent [2]. Despite NCDs accounting for about 48% of mortality, the Ethiopian government's efforts have primarily targeted infectious diseases and maternal health, leaving NCDs underfinanced [2]. The National Health Accounts report indicates that 68% of NCD services were financed by out-of-pocket expenditures from households, with only about 1% covered by development assistance [3,4].

Conceptually, interventions to address NCDs are feasible and well-established, such as the World Health Organization (WHO) Package of Essential Noncommunicable Interventions and the WHO Best Buys [5,6]. There are several effective approaches, including national policies, service delivery strategies such as screening and the provision of essential medications, as well as individual behavioral interventions [7]. However, LMICs face substantial challenges in implementing these interventions effectively [8,9].

In response to the rising burden of NCDs and in alignment with global initiatives, the Ethiopian government launched its first National Strategic Action Plan (NSAP) for the Prevention and Control of NCDs in 2016 [10]. This was followed by a second NSAP (2020–2025), [11] which further outlines the country's commitment to NCD control and management. Both strategies emphasize key components such as detection, screening, treatment, and palliative care as central to the national response to NCDs. Service is provided through the public and private health sectors. The public healthcare sector in Ethiopia is organized into a three-tier system of primary, secondary, and tertiary healthcare. The primary healthcare (PHC) system is composed of a primary hospital, health centres, and health posts. Tier two, the secondary healthcare system, constitutes general hospitals, while tier three is composed of specialised hospitals [12]. NCD services have been mainly concentrated in secondary and tertiary hospitals. Currently, the government is decentralizing and integrating NCD services to the PHC level, and providing training to healthcare professionals working at health centres and primary hospitals to diagnose and manage hypertension, diabetes, and mental health. A complicated case refers to the next level of healthcare facilities for further assessment and management.

Effective management of NCDs requires the consistent availability of trained health personnel, functional diagnostic tools, essential equipment, up-to-date clinical guidelines, and a reliable supply of medicines to ensure timely and quality care. While in Ethiopia, efforts have been made to expand the number of health facilities, increase the healthcare workforce, and improve the supply of diagnostic tools and essential drugs, these efforts have not translated into proportional improvements in service readiness [13]. Although the availability of NCD-related services has increased, the overall readiness of health facilities remains inadequate when compared to both national needs and global recommendations [14].

In Ethiopia, only a few studies have been conducted on the availability and readiness of health services for NCDs [15–18]. However, these studies are limited in scope, often focusing on a specific subset of NCDs such as cancer care, diabetes or hypertension and relying primarily on descriptive analyses of earlier (2014−2016) national data. There remains a lack of detailed empirical research examining the factors associated with NCD service availability and readiness in the country. One prior study [19] assessed the influence of basic facility characteristics such as location, managing authority, and facility type. However, the previous study uses a different dataset: the 2018 Ethiopian Service Availability and Readiness Assessment (SARA) survey. Furthermore, the scope of conditions assessed differed from the current study. While the earlier study focused on diabetes, CVDs, CRDs, and cervical cancer, the current study includes all of these except cervical cancer and additionally incorporates MNS use disorders. In addition, beyond facility characteristics, physical infrastructure and service inputs; governance and management practices at the facility level—such as regular administrative meetings, the presence of quality control mechanisms, functional health management information systems (HMIS), and external supervision—may significantly influence a facility's readiness to provide NCD services [20,21]. Therefore, in the current study, we use the most recent and comprehensive Service Provision Assessment (SPA 2021−22) survey data to assess the readiness of Ethiopia's health system to deliver NCD-related services [22]. In addition, this study extends previous research by examining health system and governance-related factors and incorporating MNS use disorder services readiness, an important disease area that has often been overlooked in earlier studies.

Having up-to-date and comprehensive information regarding the NCD service readiness of health facilities in Ethiopia is essential for formulating evidence-informed health policies and allocating resources appropriately. This study aims to fill a knowledge gap about NCD service readiness using the most recent SPA data.

## Materials and methods

### Study design and setting

A nationally representative cross-sectional survey involving randomly selected health facilities was conducted in Ethiopia between August 2021 and February 2022.

The Ethiopian healthcare system is a three-tiered, service-delivery structure, with primary-, secondary-, and tertiary-level health care. The health care tier system includes both public and private health facilities. The primary health care

system is composed of a primary hospital, health centres, and five satellite health posts constitute the Primary Health Care Unit (PHCU). Health posts are staffed with health extension workers, Ethiopian community health workers, who mainly provide essential promotional and preventive services, with limited involvement in curative services. Health centres provide both preventive and curative services and serve as referral centres and practical training sites for health extension workers. Primary hospitals offer inpatient and ambulatory services to about 100,000 people and provide emergency surgery (including caesarean sections and blood transfusions). The secondary health care system consists of general hospitals and is supposed to provide similar services to those of primary hospitals and serve, on average, one million people. They are referral centres for primary hospitals. At the third tier is the highest or tertiary health care system, consisting of specialized hospitals that serve as a referral center for general hospitals and as training centers for medical doctors and specialists. Furthermore, private health care facilities at different levels supplement the overall health care delivery.

## Population and sample

The 2021–22 Ethiopia Service Provision Assessment (ESPA) survey is designed to provide representative results for each of Ethiopia's 11 regions separately, for all facilities together, and by facility type at the national level. The total sample frame was 25,752, and of this, 1,407 facilities were selected for the survey [22]. The ESPA sample was a stratified random sample of health facilities, selected with equal probability systematic sampling. Stratification was achieved by first separating the health facilities in each region by facility type. Then, all the clinics in each region were further stratified by clinic designation (higher, medium, lower clinics, or speciality clinics). The sample allocation featured a power allocation across regions to achieve comparable survey precision. Because of their importance and their small numbers, all 413 hospitals (including government hospitals and private hospitals) were included in the survey. A representative sample of health centres, health posts, and clinics was selected and included in the survey. By facility type, health centres are slightly oversampled compared to clinics, and clinics are slightly oversampled compared to health posts. For clinics, all higher clinics were included, medium clinics were slightly oversampled compared to lower clinics, and lower clinics were slightly oversampled compared to other clinics. This oversampling strategy prioritises health facilities that play an important role in the health system, thereby increasing survey precision.

Pharmacies, diagnostic centres, regional laboratories, and individual doctors' offices were not included in the survey. In addition, some facilities were not covered in this survey because they were closed or not yet operational (7%) or for security reasons (9%). Two percent of facilities were not interviewed for a variety of reasons, including the facility converted into a COVID centre. Data were successfully collected for a total of 1,158 facilities, representing 82% of those on the original sample list. The Tigray region sample (144 facilities) was excluded from the survey sample due to security issues. The total sample, however, to assess accurately the readiness of health facilities for the management of NCDs, excluded 257 health posts that did not provide NCD services, was 901 (unweighted) health facilities.

## Data source

The current study used data from the 2021–22 ESPA survey, which was designed to collect information about the availability and delivery of healthcare services in Ethiopia and to examine the readiness of facilities to provide quality health services. The survey assessed the presence and function of components essential for the delivery of quality service for all aspects of healthcare, including the diagnosis and management of NCDs. This survey was undertaken by the Ethiopian Public Health Institute (EPHI). Technical support for the survey was provided by ICF International under the DHS Program. The survey was funded by the US Agency for International Development and the World Bank.

## Data collection

The 2021–2022 ESPA survey used four main instruments: a facility inventory questionnaire, a health provider interview questionnaire, observation protocols and an exit interview. For this study, we have used the data from 'facility inventory

questionnaire' and 'health provider questionnaire'. The instruments were developed to collect information on the availability of various health services in Ethiopia; to what extent are facilities prepared to provide each of the priority services, including information on the availability of specific items, the location and functional status of the facility, components of support systems and facility infrastructure. Data collection was performed in a computer-assisted personal interviewing (CAPI) program. The survey contains high-priority health services, all interrelated to some extent, that were assessed: child health, family planning, maternal health, and specific infectious diseases (STIs, HIV/AIDS, TB, and malaria) and NCDs. The NCD component assessed the availability and readiness of services for diabetes, CVDs, CRDs, MNS use disorders, cancer diseases, and chronic renal diseases.

## Quality assurance

Training was given to data collectors and supervisors. The main training took place from 7 July to 4 August 2021, with 23 master trainers conducting the main training. Thirty-seven team leaders and 148 interviewers, mostly health providers (nurses, nurse midwives, and clinicians), were trained as interviewers in the application of the questionnaires and computer programmes. The questionnaires were pretested to detect possible problems in the flow of the questionnaires, to gauge the length of time required to carry out the interviews, and to identify errors in the translations. The pre-test took place in health facilities in the Oromia region, around Adama City, that were not sampled in the main survey, for 2 days to test and refine the survey instruments and the computer programmes. Fieldwork supervision was done. The technical working group (TWG) members, Ministry of Health staff, Ethiopian Public Health Institute staff, and ICF personnel participated in supportive fieldwork supervision.

## Variables and measurement

**Outcome variables.** The outcome variables of the analysis is NCD-related service availability and readiness. The variables for service availability and readiness of HFs were selected based on the WHO SARA manual [23]. This study focuses on NCDs included in the ESPA survey: namely CVDs, diabetes, CRD and MNS use disorders due to their growing public health importance and data availability. These conditions aligns with recommendations from the Disease Control Priorities Project and related publications on priority interventions for NCDs and are also prioritised in the Ethiopian Ministry of Health National Action Plan for NCD Prevention and Management [11]. Cancer disease and chronic renal disease were excluded because of a lack of appropriate data on tracer items. Out of the four conditions included, we categorized diabetes, CVD and CRD as major NCDs, which is in line with global understanding of the four main types of NCDs [1].

Firstly, the disease-specific services (service availability) were evaluated by calculating the percentage proportion of facilities that provided diagnosis and/or management services for each condition separately. Services were considered available when the providers in the facility made diagnoses, prescribed treatments, or managed patients with specific NCDs. Next, the service readiness of facilities to provide specific NCD services was assessed based on the availability of predefined tracer items for service domains: trained staff and guidelines, basic equipment, basic diagnostics, and basic medicines and commodities. The list of tracer items for each domain related to diabetes, CVDs, CRDs, and MNS use disorders, as well as the process used to calculate the readiness score, was in accordance with the WHO SARA manual [23] and the detail is presented in online Supplementary S1 Table. The availability of tracer items is measured based on the observation of each tracer item by the interviewer. We defined service readiness as having functional equipment and unexpired medicines in stock to provide the NCD services. We considered equipment functional if it was operational and present in the facilities on the day of the survey. The items in each domain were re-coded as binary variables, taking the value '1' for the presence of the item and '0' for the absence of the item in the facility. To compute the mean score for each domain, the sum of the scores for each item was divided by the number of items, and the result was multiplied by 100. Each domain included in the score calculation contributes equally to the overall readiness score. The average score

from the domains was the readiness score. Those facilities with a service-specific readiness score of 70% or higher were considered to be "ready". The 70% service readiness cut-off is a commonly used benchmark in health systems research to determine whether a health facility is considered "ready" to provide a particular service [24,25].

**Explanatory variables.** The explanatory variables used in this study were facility location, facility level, managing authority (facility owned by public or private), external supervision (supervision of health facilities by federal, regional or district level with in the past six months), quality control (i.e., any quality assurance activities carried out during the past year), availability of a functional health management information system (HMIS) unit, and routine administrative meetings. The selection of these variables was based on previous studies [19,20,24].

## Data analysis

Data cleaning was performed before analysis by calculating frequencies and sorting. The sampling scheme was self-weighting within strata (region and facility type), but the probability of being sampled differed between strata; hence, sampling weights were computed for each stratum. The probability of facilities being sampled was calculated as the number of facilities in the sample divided by the total number of facilities in the stratum. Thus, weights were calculated as the inverse of the probability of being sampled and assigned to the tracer items indicator variables in the final dataset for analyses. The sampling weight was then normalised at the national level to get the health facility standard weight. The normalisation of the sampling weight was aimed at getting the total number of unweighted cases equal to the total number of weighted cases at the national level. Descriptive statistics were performed, and we summarized continuous variables with mean, standard deviation (SD), median and interquartile (Q1, Q3), whereas categorical variables were summarized with frequency and percent (%). Then we employed univariable and multivariable weighted logistic regression analysis to determine the association of the readiness of HFs to diabetes, CVDs, CRDs and MNS use disorder-related services with independent variables. All explanatory variables from univariable logistic regression that showed an association with outcome variables at $p < 0.2$ were eligible for inclusion in and multivariable logistic regression. The results of regression analysis are presented as adjusted odds ratio (AOR) with 95% confidence interval (CI) and a p-value of less than 0.05 is considered statistically significant. All analyses were conducted using STATA version 17 (StataCorp LP, College Station, TX, USA).

## Ethical considerations

The ethical approval and permission to access the data were obtained from the MEASURE DHS (available from https://www.dhsprogram.com/Data/: accessed on January 05, 2024) after a brief study concept was submitted. The IRB-approved procedures for DHS public-use datasets do not in any way allow respondents, households, or sample communities to be identified. There are no names of individuals or household addresses in the data files.

## Results

### Service availability by health facility characteristics

Out of 403 health facilities (excluding health posts), 338 (84%) offered at least one NCD service. Among the 205 public facilities, nearly 93% provided at least one NCD service, compared to 74% of the 198 private facilities (Table 1). As illustrated in Fig 1, approximately one-quarter (26%) of all facilities offered mental health services. Among facilities providing NCD services, 20% offered all four key NCD services, while nearly 70% provided services for major NCDs—diabetes, CVD, and CRD.

### Facility service-specific readiness

Table 2 shows the percentage of service-specific domains and overall service-specific readiness scores for each disease.

**Table 1. Weighted percentage of health facilities offering NCD services, by background characteristics, Ethiopia.**

| Characteristics | Number of facilities (n = 403) | Percentage of facilities that offer services for NCD | |
| --- | --- | --- | --- |
| | | At least one NCD (n = 338) | Major NCD (n = 338)* |
| **Facility type** | | | |
| Referral Hospital | 2 | 96.88 | 93.75 |
| General Hospital | 7 | 98.37 | 97.56 |
| Primary Hospital | 15 | 98.80 | 97.61 |
| Health Center | 182 | 93.74 | 75.72 |
| Specialty/Higher clinic | 8 | 96.52 | 83.72 |
| Medium clinic | 92 | 89.45 | 81.72 |
| Lower Clinic | 97 | 55.88 | 40.31 |
| **Managing authority** | | | |
| Public | 205 | 93.34 | 77.21 |
| Private | 198 | 74.26 | 61.83 |
| **Location** | | | |
| Urban | 212 | 85.97 | 76.16 |
| Rural | 191 | 81.72 | 62.42 |
| **Region** | | | |
| Afar | 7 | 82.85 | 73.08 |
| Amhara | 94 | 83.33 | 75.88 |
| Oromia | 152 | 89.68 | 76.19 |
| Somali | 15 | 87.58 | 75.17 |
| Benishangul Gumuz | 7 | 52.69 | 39.05 |
| SNNP | 68 | 69.94 | 43.45 |
| Sidama | 13 | 97.43 | 77.16 |
| Gambela | 9 | 66.37 | 24.69 |
| Harari | 2 | 80.00 | 66.67 |
| Addis Ababa | 33 | 92.33 | 85.82 |
| Dire Dawa | 3 | 91.41 | 91.41 |
| **National** | 403 | 83.95 | 69.64 |

*Major NCDs: CVDs, Diabetes and CRDs.

NCD: Non-communicable disease.

SNNP: Southern Nations, Nationalities, and Peoples.

The overall median services readiness index for diabetes was 50.0 (Q1, Q3: 33.3–62.5), indicating that half of the facilities scored below this level. Based on the 70% threshold, only 15.68% of health facilities (95% CI: 12.4%–19.7%) were considered adequately ready to provide diabetes-specific services. Notably, a significant difference was observed between public and private facilities, with 12.75% and 2.93%, respectively, meeting the adequacy benchmark. The mean domain scores for the availability of guidelines and trained providers, and essential medicines and commodities related to diabetes services were all below one-third of the maximum score of 100. Except for basic equipment, public facilities reported higher availability of guidelines and trained providers, diagnostics, and essential medicines compared to private facilities (Table 2).

CVD-related service readiness had a median score of 41.66 (Q1, Q3: 31.66–71.66). The mean domain score for the availability of basic equipment was 72.03 (SD: 18.48). Private facilities had a higher average score in basic equipment availability (78.39) than public facilities (67.53). However, private facilities scored lower across other service readiness

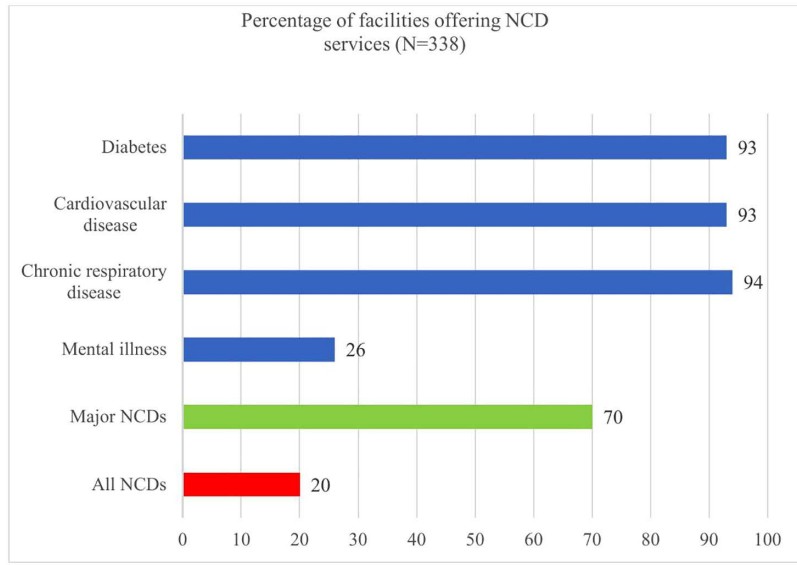

**Fig 1. Percentage of health facilities offering NCD services.**

domains. Notably, the availability of medicines and commodities was significantly lower in private facilities. Overall, only 10.64% of facilities (95% CI: 8.12%–13.82%) were considered adequately ready to provide CVD-specific services, with readiness particularly low among private facilities (2.20%).

Although CRD-related services were somewhat more available across facilities compared to diabetes and CVD, the overall median availability of tracer items was low at 31.66 (Q1, Q3: 21.66–45.00). This indicates that half of the facilities scored below 31.66, and 75% scored below 45.00, highlighting a generally low level of readiness. This is further compounded by the fact that only 3.14% (95% CI: 1.96, 5.00) of facilities were adequately prepared to provide CRD services.

Finally, the readiness of MH services was assessed across two key domains: the availability of guidelines and trained providers, and essential medicines and commodities. The overall median MNS use disorder-related service readiness score was 25.00 (Q1, Q3: 5.55–47.22), indicating that half of the facilities scored below this threshold. Public facilities generally had better availability of tracer items and higher readiness scores compared to private facilities. Based on the 70% adequacy threshold, only 0.41% of private health facilities were considered adequately ready to provide MNS use disorder-specific services. The median readiness score among private facilities was 0.00 (Q1, Q3: 0.00%–25.00%), suggesting that more than half of these facilities lacked any guidelines and trained staff, or essential medicines for MNS use disorders service delivery (Table 2).

**Factors associated with facility service readiness**

Table 3 shows the multivariable logistic regression for diabetes, CVDs, CRDs and MNS use disorder-specific service readiness index by, facility level, managing authority, location, region, administration meeting, quality control, HMIS unit and external supervision.

After adjusting for facility background characteristics, the analysis revealed that secondary and tertiary level health facilities were at least twice as likely to be "ready" to provide services for diabetes, CVDs, CRDs, and MNS use disorders compared to primary-level facilities. Health facilities located in urban areas demonstrated significantly higher readiness than their rural counterparts. Specifically, urban health facilities were 2.42 times more likely to be ready to provide diabetes services (95% CI: 1.10–5.31; $p < 0.05$), 13.66 times more likely for CRDs (95% CI: 5.10–36.58; $p < 0.001$), and 10.85

**Table 2. Service-specific percentage readiness and mean domain readiness scores for health facilities that offered NCD services.**

| Domain | Total facilities | Public facilities | Private facilities |
|---|---|---|---|
| | Score Mean (SD) | Score Mean (SD) | Score Mean (SD) |
| **Diabetes (n=314)** | | | |
| Guidelines and training | 31.41 (33.16) | 34.12 (39.81) | 28.22 (25.01) |
| Basic Equipment | 77.66 (26.06) | 74.01 (33.58) | 81.95 (16.41) |
| Basic diagnostic | 63.34 (42.39) | 71.82 (40.05) | 53.38 (39.73) |
| Essential medicine and commodities | 25.52 (30.32) | 38.89 (33.23) | 9.82 (18.89) |
| Overall readiness (Median (Q1, Q3)) | 50.00 (33.33, 62.50) | 54.16 (39.58, 68.75) | 45.83 (16.66, 62.50) |
| HFs readiness score >70 (% (95% CI)) | 15.68 (12.35, 19.70) | 12.75 (9.79, 16.44) | 2.93 (1.63, 5.20) |
| **CVDs (n=314)** | | | |
| Guidelines and training | 30.06 (33.77) | 31.85 (38.16) | 27.53 (27.25) |
| Basic Equipment | 72.03 (18.48) | 67.53 (22.33) | 78.39 (10.86) |
| Essential medicine and commodities | 36.54 (34.16) | 54.05 (28.83) | 11.78 (23.99) |
| Overall readiness (Median (Q1, Q3)) | 41.66 (31.66, 71.66) | 48.33 (36.66, 63.33) | 41.66 (25.00, 45.00) |
| HFs readiness score >70 (% (95% CI)) | 10.64 (8.12, 13.82) | 8.44 (6.28, 11.25) | 2.20 (1.13, 4.25) |
| **CRDs (n=316)** | | | |
| Guidelines and training | 30.49 (33.75) | 33.64 (38.59) | 26.32 (26.76) |
| Basic Equipment | 30.53 (16.27) | 28.31 (15) | 33.48 (15.33) |
| Essential medicine and commodities | 40.00 (33.31) | 58.27 (27.87) | 15.82 (22.58) |
| Overall readiness (Median (Q1, Q3)) | 31.66 (21.66, 45.00) | 38.33 (28.33, 51.66) | 25.00 (8.33, 31.66) |
| HFs readiness score >70 (% (95% CI)) | 3.14 (1.96, 5.00) | 2.05 (1.16, 3.59) | 1.09 (0.47, 2.49) |
| **MNS (n=88)** | | | |
| Guidelines and training | 32.64 (35.81) | 41.31 (41.27) | 14.16 (16.92) |
| Essential medicine and commodities | 29.13 (28.47) | 37.57 (30.49) | 11.13 (15.19) |
| Overall readiness (Median (Q1, Q3)) | 25.00 (5.55, 47.22) | 36.11 (16.66, 55.55) | 0.00 (0.00, 25.00) |
| HFs readiness score >70 (% (95% CI)) | 11.52 (7.48, 17.34) | 11.11 (7.12, 16.94) | 0.41 (0.18, 0.93) |

CI Confidence interval, CRD Chronic respiratory disease, CVD Cardiovascular diseases, HFs Health facilities, MNS Mental, neurological and substance, SD Standard deviation.

times more likely for MNS use disorder-related services (95% CI: 4.05–29.09; $p < 0.001$). Private facilities were notably less prepared to deliver MH services, with an AOR of 0.07 (95% CI: 0.02–0.24; $p < 0.001$), indicating they were 93% less likely to meet readiness criteria compared to public facilities. Regional disparities were also evident. Facilities in Oromia were significantly less likely to be ready for diabetes and CVD service provision. Conversely, facilities in the Southern Nations, Nationalities, and Peoples' (SNNP) region were 87% less likely to be ready for CVD services (AOR=0.13; 95% CI: 0.03–0.49; $p < 0.01$), while facilities in Sidama were 5.23 times more likely to be ready for CRD services compared to those in Addis Ababa (95% CI: 1.59–17.13; $p < 0.01$). Facility-level management practices also showed significant associations with service readiness. Facilities that held routine administrative meetings were substantially more likely to be ready to provide CVD and CRD services, with AORs of 9.76 and 13.38, respectively. Similarly, facilities with a functional HMIS unit were 4.74 times more likely to be ready to deliver CVD services. Finally, external supervision from higher or immediate administrative levels was a significant predictor of service readiness. Facilities that received external supervision were 8.02 times more likely to be ready for diabetes services and 14.39 times more likely for CVD services, compared to those that did not receive such supervision.

**Table 3. Factors associated with the readiness of health facilities to provide NCD services.**

| Variables | Diabetes service readiness (n=314) Model-1 | CVD service readiness (n=314) Model-2 | CRD service readiness (n=316) Model-3 | MNS service readiness (n=88) Model-4 |
|---|---|---|---|---|
| | AOR (95% CI) | AOR (95% CI) | AOR (95% CI) | AOR (95% CI) |
| **Facility level** | | | | |
| Primary[a] | – | – | – | – |
| Secondary[b] | 10.79 (4.86, 23.96)*** | 9.97 (4.52,21.98)*** | 4.38 (1.66, 11.54)** | 2.48 (1.02, 6.05)* |
| Tertiary[c] | 7.78 (2.29, 26.37)** | 6.78 (2.26, 20.30)** | 5.68 (1.82, 17.66)** | 4.34 (1.43, 13.15)* |
| **Managing authority** | | | | |
| Public | – | – | | – |
| Private | 0.38 (0.13, 1.04) | 0.36 (0.10, 1.30) | | 0.07 (0.02, 0.24)*** |
| **Location** | | | | |
| Rural | – | – | – | – |
| Urban | 2.42 (1.10, 5.31)* | 2.24 (0.93, 5.38) | 13.66 (5.10,36.58)*** | 10.85 (4.05, 29.09)*** |
| **Region** | | | | |
| Addis Ababa | – | – | – | |
| Afar | 0.50 (0.10, 2.43) | 0.52 (0.10, 2.51) | 0.55 (0.08, 3.63) | |
| Amhara | 1.41 (0.46, 4.32) | 0.77 (0.24, 2.51) | 1.28 (0.33, 4.92) | |
| Oromia | 0.17 (0.04, 0.06)* | 0.10 (0.02, 0.40)** | 0.64 (0.11, 3.68) | |
| Somali | 0.61 (0.13, 2.92) | 0.91 (0.18, 4.55) | 2.86 (0.64, 12.65) | |
| Benishangul Gumuz | 1.33 (0.27, 6.52) | 2.45 (0.50, 12.05) | 2.47 (0.31, 19.82) | |
| SNNP | 0.64 (0.20, 2.09) | 0.13 (0.03, 0.49)** | 0.12 (0.06, 0.70) | |
| Sidama | 0.55 (0.16, 1.82) | 0.63 (0.19, 2.08) | 5.23 (1.59, 17.13)** | |
| Gambela | 0.71 (0.14, 3.38) | 0.20 (0.01, 2.22) | Empty | |
| Harari | 1.02 (0.18, 5.67) | 0.80 (0.15, 4.29) | 0.54 (0.03, 7.89) | |
| Dire Dawa | 2.51 (0.73, 8.59) | 1.70 (0.54, 5.36) | 0.25 (0.03, 2.07) | |
| **Administration meeting** | | | | |
| No | – | – | – | |
| Yes | 2.65 (0.52, 13.46) | 9.76 (2.84, 33.54)*** | 13.38 (1.37, 129.85)* | |
| **Quality control** | | | | |
| No | – | – | – | – |
| Yes | 3.20 (1.04, 9.87)* | 1.44 (0.44, 4.64) | 14.60 (5.34, 39.96)*** | 7.48 (1.94, 28.81)** |
| **HMIS unit** | | | | |
| No | – | – | | – |
| Yes | 1.91 (0.46, 7.94) | 4.74 (1.75, 12.85)** | | 2.45 (0.17, 34.59) |
| **External Supervision** | | | | |
| No | – | – | – | |
| Yes | 8.02 (1.66, 38.60)** | 14.39 (1.74, 118.73)* | 2.22 (0.21, 22.98) | |

**Multivariable analysis adjusted for facility background characteristics:** Model-1 and Model-2 (facility level, managing authority, location, region, administration meeting, quality control, HMIS unit and external supervision); Model-3 (facility level, location, region, administration meeting, quality control, and external supervision); Model-4 (facility level, managing authority, location, quality control, and HMIS unit).

AOR Adjusted odds ratio, CI Confidence interval, HMIS Health management information system, CVD Cardiovascular diseases, CRD Chronic respiratory disease, MNS Mental, neurological and substance use.

[a]Primary level: Primary hospital, Health center, Specialty/Higher clinic, Medium clinic, Lower clinic.

[b]Secondary level: General hospital.

[c]Tertiary level: Referral hospital.

*Variables significant with p-value ≤0.05, ** Variables significant with p-value ≤0.01, ***Variables significant with p-value ≤0.0001.

## Discussion

This study investigates the availability and readiness of health facilities in Ethiopia for diagnose and manage of NCDs services such as diabetes, CVDs, CRDs, and MNS use disorders. Two aspects of NCD-specific services were evaluated: service availability, service readiness, and the factors associated with the readiness of health facilities to provide NCD care. Readiness was assessed based on the availability of predefined tracer items for service domains: trained staff and guidelines, basic equipment, basic diagnostics, and essential medicines and commodities. The findings show that the majority of health facilities (83.95%) offer at least one NCD service. However, most of them demonstrated inadequate readiness to manage these conditions across all four disease categories, indicating a lack of standard treatment guidelines, adequately trained personnel, essential equipment, and necessary medications for effective NCD care and management. Public health facilities were generally better prepared to provide NCD-related services than private facilities. In addition, facilities operated in urban areas were more likely to be prepared to provide NCD services.

In the current study, the majority of health facilities (83.95%) reported offering at least one of the four NCD services. Among these, approximately one-fifth (20%) provide all four NCD services. This represents a prominent improvement in service availability compared to the study conducted in Ethiopia using 2018 SARA survey data, where only 67.9% of facilities offered at least one NCD service and just 8% provided all four services [19]. This may result from the expansion of NCD services, mainly rolled out into primary health facilities as part of a priority action under the National Strategic Plan for the Prevention and Control of major NCDs (2020–2025) [11]. This strategic initiative aims to strengthen the health system's preparedness to manage NCDs. Among the four conditions assessed, three of them (diabetes, CVDs, and CRDs), often referred to as major NCDs, were available in approximately 93% of the facilities. However, there was a substantial disparity in the availability of services for MNS use disorder, with only 26% of facilities offering these services, highlighting a critical service gap compared to the other NCD categories. This level of service availability is below the annual target of 35% set for 2021 in Ethiopia's National Mental Health Strategy (2020–2025) [12]. Overall, the availability of NCD services in this study is higher compared to findings from similar studies conducted in other African and South Asian countries [20,25–27].

One of the key pillars of the national NCD strategic plan is strengthening health service delivery through the development of human resources and improvements in infrastructure, diagnostics, medical supplies, and technologies. These interventions are intended to build the capacity of the health system—particularly at the primary care level—to provide timely, effective, and comprehensive NCD-related services. While there has been progress compared to the 2018 SARA survey, the current findings indicate that the overall readiness of health facilities remains significantly below the 70% service readiness threshold. Only 15.68% of facilities met the readiness threshold for diabetes, 10.64% for CVDs, 3.14% for CRDs, and 11.52% for MNS use disorder-related services. Notably, although CVD is the most prevalent NCD, facilities were less prepared to manage it than diabetes. There is therefore a need for services to be prioritised according to disease burden.

The median overall readiness scores for diabetes, CVDs, CRDs, and MNS use disorders were 50.00, 41.66, 31.66, and 25.00, respectively. These median values indicate a generally low level of preparedness across all conditions, with at least 50% of facilities scoring below these levels. This suggests that the majority of facilities lack essential components such as trained personnel, clinical guidelines, diagnostic equipment, and essential medicines needed to manage NCDs effectively. Such low levels of readiness highlight critical gaps in the health system's capacity to respond to the growing burden of NCDs. A study on the integration of mental health and substance abuse services into primary health care in Ethiopia identified delayed support and supervision, high staff turnover and interruptions in the supply of essential medicines, among other key challenges [28]. Compared to other studies, our findings indicate a higher level of service readiness than reported in Nepal [26], but lower than that observed in Kenya, Vietnam, Bangladesh, and Tanzania [20,25,27,29]. These variations may be attributed to differences in health system structures, the tools used for data collection, and the extent to which NCD prevention and control are prioritised within national policies. For instance, the Kenyan study employed the

WHO-PEN tool, which is specifically designed to monitor NCD service implementation at the primary healthcare level. When disaggregating service readiness by facility ownership, public facilities were generally more prepared than private ones across all four NCD conditions. This may be attributed to their integration into national health strategies, better access to public resources, provision of more comprehensive care, and stronger accountability mechanisms.

At the domain level, the mean percentage scores for guidelines and trained staff, and essential medicines were consistently low across all conditions, with values falling below nearly one-third of the maximum score of 100. In contrast, the mean availability score for basic equipment was relatively higher, particularly for diabetes, CVDs, and CRDs. A notable finding is that, except for MNS use disorders, which were assessed using only two domains (guidelines and trained staff, and essential medicines), private facilities had relatively better availability of basic equipment, while public facilities scored higher in the availability of guidelines and trained staff, and essential medicines. Furthermore, the readiness of private facilities to deliver MH-related services was extremely poor. Almost none of the private facilities (only 0.41%) met the 70% readiness threshold for MHservices. Notably, the median overall readiness score was 0.00 (Q1, Q3: 0.00, 25.00), indicating that 75% of these facilities completely lacked trace items for guidelines and trained staff, as well as essential medicines, and even the top 25% of performers fell short of the 70% cut-off. This highlights a severe gap in the capacity of private facilities to manage MNS use disorders. This may be explained by regulatory restrictions in Ethiopia, where private clinics (primary, medium, higher, and speciality) are prohibited from holding or dispensing non-emergency medicines in these health facilities at any time [30,31]. As a result, private facilities are less likely to stock essential NCD medicines and impact their medication domain of readiness. Moreover, private facilities are poorly integrated into the health systems. This leads to poor access to clinical guidelines and training related to health services, including NCD management. Short-term and refreshment training for private facilities is provided by the government through public–private partnership (PPP) programs [12]. These partnerships are collaborative arrangements between the government and private sector actors aimed at addressing health issues. The observed differences in capacity between public and private facilities may, in part, be attributed to the government's stronger emphasis on supporting the public sector, alongside existing gaps in the implementation of effective PPP. A case study on the PPP revealed gaps that the federal and regional governments have not yet addressed in their efforts to effectively implement PPP policies through private sector engagement endeavors. The regulatory standards, by the Ethiopian Food and Drug Administration, in place were considered too stringent and not conducive to service delivery by private health facilities [32]. Another study on private sector assessment revealed that the overall regulatory framework contains many barriers to growing and harnessing the private health sector. Market conditions present the biggest constraint to private sector investment and growth and limit their ability to prepare for the service provision. Furthermore, there are limited economic and financial incentives to provide NCD services. Limited private sector understanding of government priorities. The private sector representatives interviewed stressed that they have little access to information from the MOH [33].

Our study revealed that secondary and tertiary-level health facilities— hospitals—were more likely to be ready to provide NCD-related services compared to primary-level facilities, such as health centers and private clinics. This finding aligns with similar studies conducted in Zambia, Kenya, and Nepal [24,25,34]. Except for CVD services, facilities located in urban areas were generally more prepared to deliver NCD care than those in rural areas, a pattern also observed in other studies [24,25]. Additionally, health system and management practice-related factors were significantly associated with service readiness. Facilities that received external supervision from higher administrative levels—such as district, regional, or ministerial offices—demonstrated significantly higher readiness scores for diabetes and CRDs. This is consistent with findings from studies in Uganda and Nepal [24,35]. Furthermore, facilities with functional quality control units and regular administrative meetings were more likely to be adequately prepared to provide NCD services.

## Implications for healthcare system and policy

Despite the rollout of the national NCD strategic plan [11] and the notable expansion in the availability of services, this study reveals that overall service readiness for NCDs remains significantly low across health facilities in Ethiopia. This

indicates a gap between policy ambition and implementation capacity. The marked disparity in the availability of essential medicines between public and private facilities suggests the need for healthcare facility standard reform, targeted interventions to strengthen pharmaceutical supply systems, particularly in the lower-level private facilities. This has been witnessed public–private mix (PPM) approach in the provision of TB, HIV, and malaria services verified increased service demand and utilization [32]. This was facilitated by supportive policies that enabled private facilities to access trained health workers, essential medicines, and technical support from the government. Moreover, the readiness to provide MH services is alarmingly low and heavily skewed toward public facilities, highlighting a critical service gap that requires a programmatic and aggressive response. Addressing these inequities will require a coordinated response involving strengthened PPP, regulatory reforms, increased investment in health workforce training, and improved infrastructure and supply chains, especially at the primary healthcare level.

These findings underscore the need for Ethiopia's health system to move beyond service availability and focus on enhancing the quality and preparedness of care delivery, with particular emphasis on equity, integration, and disease burden-driven resource allocation.

## Strengths and limitations

A key strength of this study is the use of a nationally representative and comprehensive dataset, ensuring national representativeness in terms of general health facilities' characteristics within the country's health system. The survey employed an objective facility assessment approach, whereby the availability and functionality of tracer items within each domain were verified through direct observation, enhancing the reliability of the findings. Another strength lies in the use of an adapted DHS Service Provision Assessment survey tool, which is widely recognized for its robustness and validity in health systems research. The application of this tool ensures a high degree of accuracy in characterizing the current service readiness landscape. Moreover, the SPA tool is increasingly being used in other LMICs, thereby enhancing the comparability of our results with similar studies conducted elsewhere. Additionally, the estimates presented in this study were adjusted and weighted to account for non-response and disproportionate sampling, further improving the representativeness and precision of the findings.

The main limitation of this study is its cross-sectional design, which restricts the ability to draw causal inferences. Additionally, the study focused predominantly on the supply side of the health system, with limited attention to demand-side factors, such as health-seeking behaviours, financial barriers, and community-level perceptions, that are also critical for informing effective interventions. While the study assessed key readiness components, including the availability of essential medicines, diagnostics, and clinical guidelines, these evaluations were largely basic. More advanced diagnostic tools and technologies required for the comprehensive management of NCDs, such as electrocardiograms and imaging equipment, were not included in the assessment. Furthermore, the study did not evaluate facility readiness for cancer and chronic kidney diseases due to the absence of data on relevant tracer items. Therefore, a primary study is needed to assess the level of facility readiness to provide these services. Finally, because of the security issue, the Tigray region was excluded from the survey sample. These gaps limit the study's ability to provide a full picture of the health system's preparedness to address the broader spectrum of NCDs.

## Conclusion

This study found that the Ethiopian health system is insufficiently prepared for CVDs, diabetes, CRDs and MHservices. The service readiness was higher in public health facilities compared to private health facilities. Given Ethiopia's commitment under Global Action Plan on NCDs, and commitments under periodic plans and policies, the country needs to strengthen service delivery platforms while improving the overall readiness of the health system through increasing the number of qualified health staff, training and provision of equipment and medicines. A number of factors have been associated with better service readiness, including the location of the facility, the level of the facility, having a quality control unit, conducting regular administrative meetings and receiving external supervision.

# Supporting information

S1 Table. Summary of tracer items of each domain and measurement procedure.
(DOCX)

# Acknowledgments

We would like to acknowledge DHS program for providing us data for further analysis. We would also like to thank Mr. Tsegaye Gebremedhin for his assistance with data analysis.

# Author contributions

**Conceptualization:** Andualem Yalew Aschalew, Jenberu Mekurianew Kelkay, Getachew Teshale, Kaleb Assegid Demissie, Nebebe Demis Baykemagn, Melak Jejaw.

**Data curation:** Andualem Yalew Aschalew, Jenberu Mekurianew Kelkay, Getachew Teshale, Kaleb Assegid Demissie, Nebebe Demis Baykemagn, Tesfahun Zemene Tafere, Asebe Hagos, Melak Jejaw.

**Formal analysis:** Andualem Yalew Aschalew, Jenberu Mekurianew Kelkay, Getachew Teshale, Kaleb Assegid Demissie, Azmeraw Tadele, Misganaw Guadie Tiruneh, Tesfahun Zemene Tafere, Asebe Hagos, Melak Jejaw.

**Investigation:** Andualem Yalew Aschalew, Kaleb Assegid Demissie.

**Methodology:** Andualem Yalew Aschalew, Jenberu Mekurianew Kelkay, Getachew Teshale, Kaleb Assegid Demissie, Nebebe Demis Baykemagn, Azmeraw Tadele, Misganaw Guadie Tiruneh, Tesfahun Zemene Tafere, Asebe Hagos, Melak Jejaw.

**Software:** Andualem Yalew Aschalew, Azmeraw Tadele, Misganaw Guadie Tiruneh, Asebe Hagos.

**Validation:** Andualem Yalew Aschalew, Kaleb Assegid Demissie.

**Visualization:** Andualem Yalew Aschalew, Melak Jejaw.

**Writing – original draft:** Andualem Yalew Aschalew.

**Writing – review & editing:** Andualem Yalew Aschalew, Jenberu Mekurianew Kelkay, Getachew Teshale, Kaleb Assegid Demissie, Nebebe Demis Baykemagn, Azmeraw Tadele, Misganaw Guadie Tiruneh, Tesfahun Zemene Tafere, Asebe Hagos, Melak Jejaw.

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
