## [Decision Letter · Decision Letter 0]

1 Jul 2025

Dear Dr. Aschalew,

Thank you for submitting your manuscript to PLOS ONE. After careful consideration, we feel that it has merit but does not fully meet PLOS ONE’s publication criteria as it currently stands. Therefore, we invite you to submit a revised version of the manuscript that addresses the points raised during the review process.

We look forward to receiving your revised manuscript.

Kind regards,

Nipun Shrestha, Ph.D.

Academic Editor

PLOS ONE

Journal Requirements:

3. In the online submission form, you indicated that all relevant data contributing to the findings are within the paper. Data used in our study are publicly available upon request from the DHS program website. (https://dhsprogram.com/ ).

4. Please include a copy of Table 3 which you refer to in your text on page 14.

5. We note you have included a table to which you do not refer in the text of your manuscript. Please ensure that you refer to Table 4 in your text; if accepted, production will need this reference to link the reader to the Table.

Additional Editor Comments :

In the discussion section authors pointed out that private health facilities are prohibited to hold or dispense non-emergency medicines in these health facilities at any time. This might a primary reason for poor preparedness of private health facilities in managing NCDs in Ethiopia. I would suggest authors to conduct a supplementary analysis to see if the private health facilities are performing poorly only due to these restrictions compared to public health facilities or there are other reasons as well.

Reviewers' comments:

Reviewer's Responses to Questions

**Comments to the Author**

1. Is the manuscript technically sound, and do the data support the conclusions?

Reviewer #1: Yes

Reviewer #2: Yes

2. Has the statistical analysis been performed appropriately and rigorously?

Reviewer #1: Yes

Reviewer #2: Yes

3. Have the authors made all data underlying the findings in their manuscript fully available?

Reviewer #1: Yes

Reviewer #2: Yes

4. Is the manuscript presented in an intelligible fashion and written in standard English?

Reviewer #1: Yes

Reviewer #2: Yes

Reviewer #1: Introduction section is well written and address the gap in literature in the context of Ethiopia.

In method section, please mention how many hospitals were included in the sample.

Line 195, What are the variable definitions of managing authority and external supervision.

Some information seem to be repetitive and can be shortened.

How is Major NCD is defined in Table 1? The categorization if different than mentioned in the first paragraph should be spelled out in the methods.

Could you please add a footnote to the table that explains what facility background characteristics were used to adjust in the multivariable analysis?

The overall readiness of private facilities is startling, and there must be a good discussion of why this is the case. Expand your discussion with some evidence in the Ethiopia’s health care system perspective.

Reviewer #2: 1. Background: Please briefly describe the current noncommunicable disease service delivery mechanism in Ethiopia

2. Methods: In the methods section, could you elaborate on the study reliability and quality assurance for data collection.

3. A strong rationale is needed on how this study will add up to the existing literature and the preceding survey findings .

4. Were there any eligibility criteria for the health facilities to be included in the study?

5. Flowing paper highlights some of the changes across the surveys to identify how availability and readiness in NCD services have changed over the years,

https://bmchealthservres.biomedcentral.com/articles/10.1186/s12913-024-11606-8

As the the authors mentioned , there was analysis for 2018 survey, this paper should highlight the changes across the surveys to identify critical factors for changes in service availability

6. Line 268 to 374 need to backed up by evidence. Can you please explain the type of public private partnership and how this supports capacity building of Public health facilities?

7. The authors need to be clearly outlined the implication of the findings and further recommendations for improving the study, or strengthening the health system

**Do you want your identity to be public for this peer review?** For information about this choice, including consent withdrawal, please see our Privacy Policy

Reviewer #1: **Yes: ** Umesh Ghimire

Reviewer #2: No

---

## [Author Response · Author response to Decision Letter 1]

27 Oct 2025

Manuscript title: Availability and readiness of health facilities for non-communicable disease-related services in Ethiopia: evidence from the nationally representative health facility survey 2022

Manuscript ID: PONE-D-25-27596

Response to Journal requirements

Comment 1:

Response:

Thank you for your invaluable comment and for sharing the links for the improvement of our manuscript. We have reformatted the manuscript according to the above style guidelines, including file naming. Please look at page one of the manuscript.

Comment 2:

Response:

Thank you for the recommendation. All relevant data contributing to the findings are within the manuscript itself, and data used in our study will be uploaded as supplementary information on acceptance.

Comment 3:

3. In the online submission form, you indicated that all relevant data contributing to the findings are within the paper. Data used in our study are publicly available upon request from the DHS program website. (https://dhsprogram.com/).

Response:

Thank you for the comments. All relevant data contributing to the findings are within the manuscript itself, and data used in our study will be uploaded as supplementary information on acceptance.

Comment 4:

4. Please include a copy of Table 3 which you refer to in your text on page 14.

Response:

We greatly appreciate your detailed observation and valuable feedback. The text “Table 3” on page 14 was correct and referred to table on page 15. There was an editorial mistake when writing the title of Table 3 on page 15. It should have been labeled as Table 3 instead of Table 4. Furthermore, there was no table labeled as Table 4 in the manuscript. We correct the editorial error. Please look at it on page 16.

Comment 5:

5. We note you have included a table to which you do not refer in the text of your manuscript. Please ensure that you refer to Table 4 in your text; if accepted, production will need this reference to link the reader to the Table.

Response:

Thank you for your comment. This has been addressed with the previous comment and please look at the response.

Comment 6:

Additional Editor Comments :

In the discussion section authors pointed out that private health facilities are prohibited to hold or dispense non-emergency medicines in these health facilities at any time. This might a primary reason for poor preparedness of private health facilities in managing NCDs in Ethiopia. I would suggest authors to conduct a supplementary analysis to see if the private health facilities are performing poorly only due to these restrictions compared to public health facilities or there are other reasons as well.

Response:

Thank you for your constructive comments. We have agreed with you that the prohibition of holding medicine was not the only reason for the low readiness score of private facilities. They have also scored lower in the Guideline and staff domain and basic diagnostics; in fact, they scored better in the equipment domain and this supplementary analysis was presented in Table 2 on page 14. In addition, in the discussion section, we elaborate more on why they have low readiness, considering the Ethiopian healthcare system and available evidence. Please look at page 20, lines 412-428.

Response to Reviewers

Reviewer #1

Comment 1:

In method section, please mention how many hospitals were included in the sample.

Response:

Dear reviewer, thank you for your comment. We have included the no of hospitals in the sample section. (page 7, line number 138).

Comment 2:

Line 195, What are the variable definitions of managing authority and external supervision.

Response:

The term “managing authority” represents who owns and runs the health facility. In our study, we have two categories: private facilities and public facilities. The other term “external supervision” stands for whether the facilities have had supervision from one of the external bodies, like federal, regional or district health supervisors/managers, in the past six months. We have included these definitions in the revised manuscript, page 10, lines 220-221.

Comment 3:

Some information seems to be repetitive and can be shortened.

Response:

We appreciate this valuable suggestion. We revised and shortened some of this repetitive information throughout the manuscript e.g years of the survey (2021-2022 ESPA), list of conditions, how readiness was calculated etc.

Comment 4:

How is Major NCD is defined in Table 1? The categorization if different than mentioned in the first paragraph should be spelled out in the methods.

Response:

Thank you for your invaluable comments. The definition in the first paragraph of the introduction was meant to provide the global burdens of the problems. So in most literature, including the WHO, the major burden of NCD is four (CVD, diabetes, cancer and CRD), and we intended to show the share of the burden among the total NCD burden. In Table 1, we put the footnote that in our study, we include only three of the four major NCDs, which are CVDs, Diabetes and CRDs, because there were no adequate tracer items for cancer from the survey data. We mentioned in the method section, lines 194-197, that we exclude cancer because of data unavailability.

Comment 5:

Could you please add a footnote to the table that explains what facility background characteristics were used to adjust in the multivariable analysis?

Response:

Thank you for the comments. We have included a footnote to Table 3 that explains the background characteristics used to adjust in the analysis.

Comment 6:

The overall readiness of private facilities is startling, and there must be a good discussion of why this is the case. Expand your discussion with some evidence in the Ethiopia’s health care system perspective.

Response:

That is a very insightful comment. In our study, the overall readiness of private facilities was lower than that of public ones. In the first submission, we provided some possible explanations, such as the legal prohibition of holding medication in private facilities. Considering your comments, we expanded our discussion after reviewing additional relevant literature, and some of the added explanations are the poor integration of private facilities into the national NCD program, inadequate implementation of the public-private partnership policy, market conditions, and limited economic and financial incentives to provide NCD services. Please see page 20, lines 412-428.

Response to Reviewers

Reviewer #2

Comment 1:

Background: Please briefly describe the current noncommunicable disease service delivery mechanism in Ethiopia

Response:

Thank you for your comments. We have included some description on how healthcare services are provided, including the structures in Ethiopia, specifically for NCDs. Please see lines 72-80.

Comment 2:

Methods: In the methods section, could you elaborate on the study reliability and quality assurance for data collection.

Response:

Thank you for this helpful suggestion. We have included quality assurance subtopics in the method section (page 9) that explain reliability and quality assurance for data collection, such as training for data collectors, pre-test for questionnaire, supervision on data collection etc.

Comment 3:

A strong rationale is needed on how this study will add up to the existing literature and the preceding survey findings .

Response:

We agree on the essentials of a strong rationale. We incorporated additional rationales about the contribution of this study to the existing literature, as the previous study in Ethiopia used a different dataset (2018), and the scope of the conditions included was different. In this study, we include mental, neurological, and substance use disorders in addition to other conditions. In addition, the previous study considered only facility characteristics in the analysis of associated factors. But in our study, we included health system and governance-related factors like health information system, administrative meeting, quality control mechanism etc. Furthermore, this study used the recent dataset (SPA 2022), which can show the potential change in NCD service availability and readiness since the second national NCD policy has implemented since 2020. Please look at lines 88-110.

Comment 4:

Were there any eligibility criteria for the health facilities to be included in the study?

Response:

That is a good question. As we mentioned in the methods, population and sample section, the survey provides a representative sample of facilities in Ethiopia. The sample was a stratified sample. Because of their importance and their small numbers, all 413 hospitals (including public hospitals and private hospitals) were included in the survey. However, a representative sample of health centres, health posts, and clinics was selected and included in the survey. In addition, pharmacies, diagnostic centres, regional laboratories, and individual doctors’ offices were not included in the survey, and some facilities were not covered in the survey because they were closed or not yet operational.

Comment 5:

Flowing paper highlights some of the changes across the surveys to identify how availability and readiness in NCD services have changed over the years,

https://bmchealthservres.biomedcentral.com/articles/10.1186/s12913-024-11606-8

As the the authors mentioned , there was analysis for 2018 survey, this paper should highlight the changes across the surveys to identify critical factors for changes in service availability

Response:

Dear reviewer, thank you for sharing such an important article. Just to highlight, the article from Nepal was an analysis of a two-point dataset, whereas our study used a point dataset. However, we appreciated the difference between the 2018 survey and the current survey (2022). One of the differences is, the 2018 survey was the Service Availability and Readiness Assessment survey and did not contain mental, neurological and substance use disorders, while our study was based on the service provision and assessment (SPA-2022) survey. In addition, the 2018 survey was a bit old, as the second national NCD policy has started to be implemented since 2020 and might lead to some changes in service availability.

Comment 6:

Line 268 to 374 need to backed up by evidence. Can you please explain the type of public private partnership and how this supports capacity building of Public health facilities?

Response:

We appreciate your comments. However, lines 268 to 374 deal with the result section, including tables and the discussion section. Table 2 is evidence showing the difference between public and private facilities' readiness scores, including the domain level and total scores. However, if you were referring to lines 368–374, we have expanded our discussion to incorporate data from the current study and potential explanations based on existing evidence from the Ethiopian context. Please look at lines 410 to 428.

Comment 7:

The authors need to be clearly outlined the implication of the findings and further recommendations for improving the study, or strengthening the health system

Response:

Thank you for the comments. We already have the implications and strengths and limitations sections. We pointed out the implications of the current study to the healthcare system, the strengths of the current study and the limitations that this study did not address and recommendations for further study. Please look at lines 440 to 482.

---

## [Editor Report · Decision Letter 1]

29 Oct 2025

Availability and readiness of health facilities for non-communicable disease-related services in Ethiopia: evidence from the nationally representative health facility survey 2022

PONE-D-25-27596R1

Dear Dr. Aschalew,

We’re pleased to inform you that your manuscript has been judged scientifically suitable for publication and will be formally accepted for publication once it meets all outstanding technical requirements.

Kind regards,

Nipun Shrestha, Ph.D.

Academic Editor

PLOS ONE

---

## [Editor Report · Acceptance letter]

PONE-D-25-27596R1

PLOS ONE

Dear Dr. Aschalew,

I'm pleased to inform you that your manuscript has been deemed suitable for publication in PLOS ONE. Congratulations! Your manuscript is now being handed over to our production team.

Kind regards,

on behalf of

Dr. Nipun Shrestha

Academic Editor

PLOS ONE